# An Improved CASA Model for Estimating Winter Wheat Yield from Remote Sensing Images

**Yulong Wang [1,2], Xingang Xu [1,*], Linsheng Huang [2], Guijun Yang [1], Lingling Fan [1,2], Pengfei Wei [1] and Guo Chen [1]**

[1]  Key Laboratory of Quantitative Remote Sensing in Agriculture of Ministry of Agriculture, Beijing Research Center for Information Technology in Agriculture, Beijing Academy of Agriculture and Forestry Sciences, Beijing 100097, China; p16201079@stu.ahu.edu.cn (Y.W.); yanggj@nercita.org.cn (G.Y.); p17301156@stu.ahu.edu.cn (L.F.); p17201081@stu.ahu.edu.cn (P.W.); p17301135@stu.ahu.edu.cn (G.C.)

[2]  National Engineering Research Center for Agro-Ecological Big Data Analysis & Application, Anhui University, Hefei 230601, China; linsheng0808@ahu.edu.cn

*  Correspondence: xxg2007@aliyun.com; Tel.: +86-010-5150-3676

**Abstract:** The accurate and timely monitoring and evaluation of the regional grain crop yield is more significant for formulating import and export plans of agricultural products, regulating grain markets and adjusting the planting structure. In this study, an improved Carnegie–Ames–Stanford approach (CASA) model was coupled with time-series satellite remote sensing images to estimate winter wheat yield. Firstly, in 2009 the entire growing season of winter wheat in the two districts of Tongzhou and Shunyi of Beijing was divided into 54 stages at five-day intervals. Net Primary Production (NPP) of winter wheat was estimated by the improved CASA model with HJ-1A/B satellite images from 39 transits. For the 15 stages without HJ-1A/B transit, MOD17A2H data products were interpolated to obtain the spatial distribution of winter wheat NPP at 5-day intervals over the entire growing season of winter wheat. Then, an NPP-yield conversion model was utilized to estimate winter wheat yield in the study area. Finally, the accuracy of the method to estimate winter wheat yield with remote sensing images was verified by comparing its results to the ground-measured yield. The results showed that the estimated yield of winter wheat based on remote sensing images is consistent with the ground-measured yield, with $R^2$ of 0.56, RMSE of 1.22 t ha$^{-1}$, and an average relative error of −6.01%. Based on time-series satellite remote sensing images, the improved CASA model can be used to estimate the NPP and thereby the yield of regional winter wheat. This approach satisfies the accuracy requirements for estimating regional winter wheat yield and thus may be used in actual applications. It also provides a technical reference for estimating large-scale crop yield.

**Keywords:** remote sensing; winter wheat; CASA model; NPP; yield

## 1. Introduction

Food is the basis of human survival and development. However, in recent years, with the increase in population and human activity, the global ecosystem has been threatened. Air pollution and the loss of cultivated land have had significant impacts on grain crops, creating a major worldwide challenge in the form of food shortages and food security [1–3]. Accurate and timely monitoring and evaluation of regional grain yield is vital to the scientific formulation of agricultural import and export plans, the regulation of the grain market, and the guidance and adjustment of planting structures [4–7]. At present, with the rapid development of remote sensing technology, the traditional time-consuming methods for estimating grain yield, such as statistical surveying, agronomy forecasting, and agrometeorology, have gradually integrated the use of remote sensing images to more rapidly estimate grain yield and other factors over large areas [8–11].

In recent years, research has progressed in the field of estimating crop yield based on remote sensing images [12–16]. In particular, research has focused on methods to calculate the net primary production (NPP) of crops based on remote sensing data to estimate the mass of crop dry matter quality and the associated crop yield [17]. The conventional models used to estimate models crop yield from remote sensing data include statistical, crop-growth, and parameter models [18]. Examples of the statistical model include the work of Xu et al. [19], who used Landsat5 TM remote sensing images as input into an iteration algorithm of weighted optimal combination (WOC) to assign the optimal weight to barley yield over multiple time periods. Lewis et al. [20] then established a more accurate statistical model to estimate yield by using the maximum Normalized Difference Vegetation Index (NDVI) during the crop-growing season as the yield-evaluation index. Kalubarme et al. [21] used a time-series NDVI curve to extract numerous characteristic parameters and constructed a model based on multiple linear regression analysis of yield to estimate the winter wheat yield. However, these statistical models for estimating yield are not based on crop physiology, which causes problems such as low stability and poor spatial and temporal expansion. In contrast, the high-precision crop-growth models: Crop Environment Resource Synthesis (CERES) [22], Soil Water Atmosphere Plant (SWAP) [23], and World Food Study (WOFOST) [24,25] take full account of the physiological process of crop formation. Unfortunately, these models are too complex and involve a large number of input parameters, so they are only suitable for small areas [8,26]. Parameter model combines the characteristics of empirical and mechanistic models and therefore offers the advantages of both types of models, such as simple structure, less complicated parameters, easy access, and strong applicability. It not only meets the requirements of national and even global large-scale NPP estimates, but can also be used on regional scales. It is a good model for estimating yield by simulating crop NPP [27–29].

A typical parameter model is the Carnegie–Ames–Stanford approach (CASA) model, which was originally developed from the model proposed by Monteith in 1972. The CASA model estimates the NPP by focusing on the driving role of the absorbed photosynthetically active radiation (APAR) and the light use efficiency ($\varepsilon$) absorbed by vegetation [30]. Since the development of the CASA model, it has been widely used to estimate the NPP [31–34]. However, the original CASA model is established based on estimates of the NPP of various types of vegetation in North America, and numerous deficiencies remain in the calculation and processing of some of its parameters. For example, the original CASA model used the maximum light use efficiency ($\varepsilon^*$) = 0.389 g C MJ$^{-1}$ as the maximum light use efficiency and did not distinguish between vegetation types. For the fraction of absorbed photosynthetically active radiation (fPAR), the original CASA model only established a linear relationship with the Ratio Vegetation Index (SR), which did not properly represent the internal relationship between fPAR and vegetation. In addition, the original CASA model used the soil-moisture model to calculate the water stress factor which was based on the light use efficiency, but the parameters of the soil-moisture model were difficult to obtain, and complex to calculate—making this approach unsuitable for application over a small region. Thus, with the change of environment and climatic conditions and plant characteristics, the original CASA model is not suitable to estimate the NPP of vegetation. It is vital to improve the original CASA model based on actual production and applications, which may be done by incorporating high-quality remote sensing data.

Given this motivation, the major objectives of this study are (1) to optimize the important parameters of the original CASA model in order to fit the growth of winter wheat in the study area; (2) to obtain the high temporal and spatial distribution of winter wheat NPP at five-day cycle intervals in the growing season based on the improved CASA model; (3) to analyze the distribution of winter wheat yield in 2009 combined with the NPP-yield conversion model. Therefore, two multispectral data: HJ-1A/B remote sensing images and the MOD17A2H data products, as data sources in this study. In addition, the accuracy of this estimation was verified by the measured yield data.

## 2. Study Area and Data

### 2.1. Study Area

The winter wheat experiment was carried out in the Tongzhou and Shunyi districts of Beijing from 2008 to 2009, as indicated in Figure 1.

Tongzhou District is located southeast of Beijing. It is 36.5 km wide from east to west and 48 km long from north to south, for an area of 906 km$^2$. The highest altitude is 27.6 m and the lowest altitude is 8.2 m. It has a typical continental monsoon climate with distinct seasons; with rain and heat in the same season, a large temperature difference between day and night, an annual average temperature of 10–12 °C, an annual average precipitation of 620 mm, and an average frost-free period of about 180 days. Winter wheat was sown over an area of 156 km$^2$.

Shunyi District is located northeast of Beijing. It is 45 km from east to west and 30 km from north to south, for a total area of 1021 km$^2$. The plain area accounts for 95.7%. The terrain is high in the north and low in the south. The highest point in the northern mountains is 637 m above sea level. It has a temperate continental semi-humid monsoon climate with distinct seasons; with cold and dry in winter, hot and rainy in summer, an annual average temperature of 11.5 °C, an annual average precipitation of 610 mm, and an average frost-free of about 195 days. The average elevation is 35 m and the planting area covers 139 km$^2$.

The main grain crops in the study area are winter wheat and summer maize. Winter wheat is the main crop of summer grain in both Tongzhou District and Shunyi District, and its sown area and harvested yield are in the forefront in Beijing.

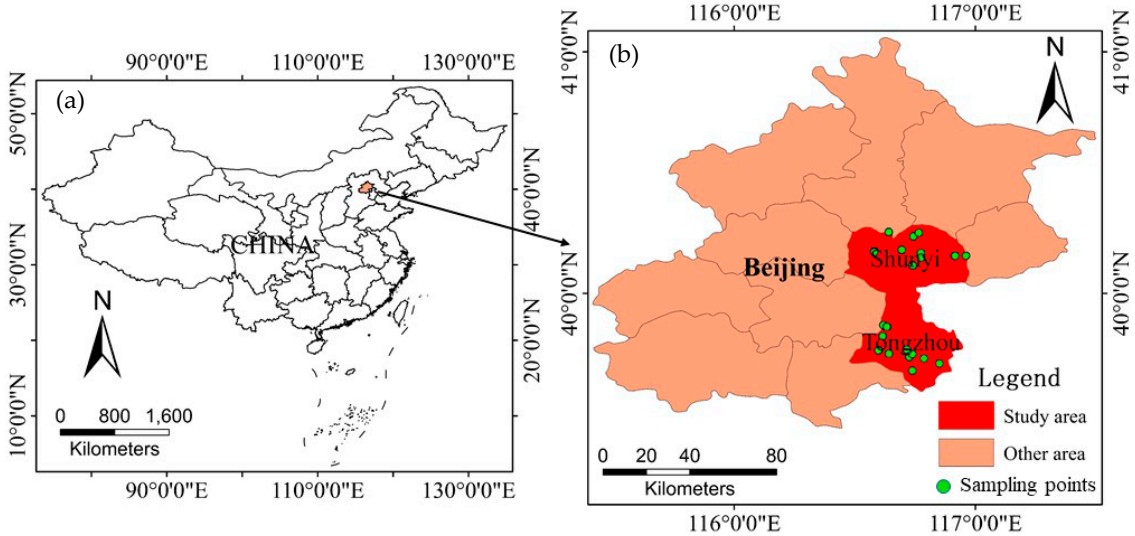

**Figure 1.** Study area: (**a**) Beijing in China; (**b**) Tongzhou and Shunyi districts in Beijing. Green dots are sampling points.

### 2.2. Data and Processing

2.2.1. Remote Sensing Data

The remote sensing data used in this study came from two sources: HJ-1A/B satellite images with a spatial resolution of 30 m and MOD17A2H data with a spatial resolution of 500 m from 2008 to 2009.

(1) HJ-1A/B Data Product

China's independently developed environment and disaster monitoring and forecasting small satellite constellations A and B, called HJ-1A/B, are equipped with CCD sensors with a spatial resolution of 30 m, radiometric resolution of 8 bit, spectral resolution (visible and near-infrared bands:

430–520 nm, 520–600 nm, 630–690 nm, and 760–900 nm). When satellites A and B are used in parallel, the return visit period can attain two days [35,36]. The level-2 products of the HJ-1A/B satellite used in this work are taken from the China Centre For Resources Satellite Data and Application website (http://www.cresda.com) [37]. From the beginning of October 2008 to the end of June 2009, high-quality images were downloaded following the principle of no cloud or less cloud in the study area. High-quality remote sensing image data for 39 scenes were screened and reserved for atmospheric radiation correction, geometric correction, image splicing, clipping, and other image preprocessing operations.

According to the growth characteristics of winter wheat, the winter wheat growing season was divided into 54 stages at 5-day cycle intervals, 39 of which could be applied to the 39 scenes of the preprocessed HJ-1A/B images and used as input data for the CASA model to extract the NDVI of internal stages. In addition, Savitzky–Golay filter was applied to eliminate image noise for the preprocessed images [38–40].

Then, a supervised classification method of maximum likelihood combined with decision tree classification was used to complete the extraction of the winter wheat planting area based on processed HJ-1A/B images. The wheat extraction results are shown in Figure 2.

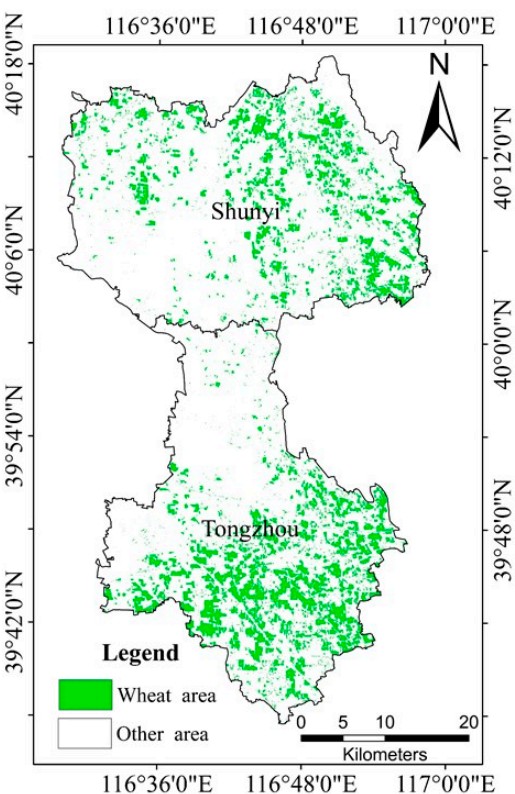

**Figure 2.** Winter wheat planting area in Tongzhou and Shunyi districts of Beijing in 2009.

(2) MOD17A2H Data Product

MOD17A2 is the primary data product of the Moderate Resolution Imaging Spectrometer (MODIS) with a spectrum range of 400–14400 nm, 36 spectral bands, and a radiometric resolution of 12-bit. It offers the first periodic, near-real-time, repetitive monitoring of global vegetation, and provides Gross Primary Production (GPP) synthesized data products with a resolution of 1 km and 8-day [41,42]. MOD17A2H is an upgrade of MOD17A2, which offers level-4 products with a spatial resolution of 500 m and temporal resolution of 8-day.

For the 15 stages of the study not covered by the HJ-1A/B data with medium spatial resolution, MODIS land standard MOD17A2H data products were used for interpolation in 2008 and 2009,

which were downloaded from the NASA website (https://ladsweb.modaps.eosdis.nasa.gov/) [43]. Its data preprocessing mainly included projection conversion, data mosaic, clipping, and so on. The MOD17A2H data products of spatial resolution were changed from 500 m to 30 m by resampling in order to extract the winter wheat images of single vegetation. In addition, the MOD17A2H value represented the daily cumulative value of GPP in 8-day, which was multiplied by $\frac{5}{8}$ can be considered as the cumulative value of GPP for five days.

Gross primary production (GPP) is the total amount of organic matter produced by photosynthesis of green plants, which is the sum of NPP and autotrophic respiration [44]. Since the NPP model was used to estimate the winter wheat yield in this work, the GPP generated by winter wheat photosynthesis was transformed into the NPP. According to numerous studies, the proportion in the GPP of organic mass consumed by autotrophic respiration during crop growth can be regarded as a fixed [45], with a constant of proportionality of 0.42 [46,47]. Therefore, the value 0.58 can be used as the conversion coefficient between GPP and NPP. Processing and calculating the selected MOD17A2H data products gave the NPP of winter wheat with a time resolution of 5-day, which was used in our model to estimate the yield of winter wheat.

### 2.2.2. Meteorological Data

The average daily temperature, hours of sunshine, and daily precipitation used herein were provided by the National Meteorological Information Center (http://data.cma.cn/) [48]. The processing consisted of the following sections:

(1) The meteorological data from five meteorological stations around the study area was selected, and then it was arranged with 5-day intervals to obtain the 5-day-average temperature, total hours of sunshine, and total precipitation.
(2) The spatial coordinate information based on the longitude and latitude of each station was assigned to meteorological data.
(3) The meteorological data with a spatial resolution of 30 m was produced by spatial interpolation of the date from the five meteorological stations.

### 2.2.3. Measured Yield Data

A total of 29 samples were selected from the study area, 15 from the Tongzhou District and 14 from the Shunyi District. The location of the sample area was determined by considering the representative growth status and regional distribution of winter wheat, and the size of each sample area was not less than 100 m × 100 m. The area of each sampling point was 1 m × 1 m. Five-point sampling was used for sampling. After drying and weighing, the average weight of winter wheat from the five sampling points was used as the yield data for the given sample area.

## 3. Study Methods

### 3.1. Construction of Improved CASA Model

Net primary production (NPP) refers to the total amount of organic matter accumulated by plants in the primary production stage per unit time and unit area. It is an essential indicator of plant-growth status [34]. The improved CASA model was used in this study to simulate the winter wheat NPP [49], and then the yield of the study area was estimated. After years of development, the parameters of the CASA model can be obtained by the inversion of remote sensing data and processing of meteorological data. This is done by using

$$\text{NPP}(x,t) = \text{APAR}(x,t) \times \varepsilon(x,t), \tag{1}$$

where $t$ is time, $x$ is the pixel number, $\text{NPP}(x,t)$ [g C m$^{-2}$] is the net primary production of pixel $x$ at time $t$, $\text{APAR}(x,t)$ [MJ m$^{-2}$] is the absorbed photosynthetically active radiation of pixel $x$ at time $t$, and $\varepsilon(x,t)$ [g C MJ$^{-1}$] is the light use efficiency for pixel $x$ at time $t$.

### 3.2. Determination of Absorbed Photosynthetically Active Radiation

The absorbed photosynthetically active radiation by actual vegetation (APAR) is mainly related to two factors: the total solar radiation SOL and the fraction of absorption by vegetation to APAR. This is calculated as follows:

$$APAR(x,t) = \frac{1}{2} \times SOL(x,t) \times fPAR(x,t), \tag{2}$$

where $SOL(x, t)$ [MJ m$^{-2}$] is the total solar radiation of pixel $x$ at time $t$, fPAR($x$, $t$) is the fraction of absorbed photosynthetically active radiation, and the factor $\frac{1}{2}$ is the ratio of effective solar radiation that can be used by vegetation to the total solar radiation.

### 3.2.1. Determination of Total Solar Radiation

The direct calculation of the total solar radiation was not included in this study because the small number of meteorological stations that provide total solar radiation. Therefore, the daily solar radiation was calculated by using the hours of sunshine, and the total solar radiation at 5-day intervals is incorporated separately by using

$$SOL(x,t) = \left(a + b\frac{n}{N}\right)R_a, \tag{3}$$

where $SOL(x, t)$ is the same as for Equation (2), and $a$ and $b$ are empirical coefficients that are determined from the measured solar radiation. In this study, the method of Zuo et al. [50] was used to calculate $a$ and $b$. For the Beijing area, $a = 0.22$ and $b = 0.72$. The ratio $\frac{n}{N}$, where $n$ is the hours of sunshine and $N = 24$ h, gives the fraction of sunshine time for one day. Finally, $R_a$ [MJ m$^{-2}$] is the solar intensity at ground level.

The quantity $R_a$ can be calculated from technical documents published by Food and Agriculture Organization of the United Nations (FAO) [51], which is related to the distance $d$ between the Sun and Earth, the latitude $\varphi$, the solar sunset angle $\omega_0$, the solar constant $S_0 = 0.082$ MJ m$^{-2}$ min, the solar declination delta $\delta$, and the number $J$ of days. $R_a$ is calculated by using

$$R_a = \frac{24*60}{\pi} \times S_0 d(\omega_0 sin\varphi sin\delta + cos\varphi cos\delta sin\omega_0), \tag{4}$$

In Equation (4), the parameters are given by

$$d = 1 + 0.033\cos\left(\frac{2\pi}{365}J\right), \tag{5}$$

$$\omega_0 = arccos(-tan\varphi tan\delta), \tag{6}$$

$$\delta = 0.409\sin\left(\frac{2\pi}{365}J - 1.39\right), \tag{7}$$

### 3.2.2. Improved Calculation of fPAR

For winter wheat, fPAR depends on the growing season and the growing period. References [18,52–54] showed the NDVI correctly represents the vegetation type and growth situation, and it is linear in fPAR. In the present study, the maximum and minimum NDVI and fPAR can be determined by HJ-1A/B images to calculate fPAR(x, t) as follows:

$$fPAR(x,t) = \frac{[NDVI(x,t) - NDVI_{min}](fPAR_{max} - fPAR_{min})}{(NDVI_{max} - NDVI_{min})} + fPAR_{min}, \tag{8}$$

In addition, Field et al. [55] found that fPAR is linear in SR:

$$\text{fPAR}(x,t) = \frac{[\text{SR}(x,t) - \text{SR}_{\min}](\text{fPAR}_{\max} - \text{fPAR}_{\min})}{(\text{SR}_{\max} - \text{SR}_{\min})} + \text{fPAR}_{\min},\tag{9}$$

$$\text{SR} = \frac{1 + \text{NDVI}(x,t)}{1 - \text{NDVI}(x,t)},\tag{10}$$

where $\text{fPAR}_{\max} = 0.950$ and $\text{fPAR}_{\min} = 0.001$ are the constant maximum and minimum values of fPAR.

The constants $\text{NDVI}_{\max}$ and $\text{NDVI}_{\min}$ are the maximum and minimum NDVI for winter wheat. The determination of NDVI maximum and minimum has always been controversial. After making a comparative analysis, the 95% lower-side value and 5% lower-side value of the winter wheat NDVI maximum and minimum probability distribution for $\text{NDVI}_{\max}$ and $\text{NDVI}_{\min}$ was used in this study, respectively [56]. However, the present study covers the entire growing season of winter wheat, which translates into 39 scenes of HJ-1A/B data and 15 scenes of MOD17A2H data, with more processed images than usual for such studies. Thus, the available HJ-1A/B images for each month with the NDVIs for winter wheat were applied, which may differ greatly between months during the growing season of winter wheat, to calculate the monthly maximum and minimum NDVI by the calculation method mentioned above. $\text{SR}_{\max}$ and $\text{SR}_{\min}$ are determined from $\text{NDVI}_{\max}$ and $\text{NDVI}_{\min}$, respectively.

With the improved CASA model proposed herein, both vegetation indices NDVI and SR can be used to estimate fPAR. According to previous research, the calculation of fPAR based on NDVI is often greater than the NVDI based on measurements, and the calculation of fPAR based on SR is often lower than the NVDI based on measurements. Therefore, the same weight $\alpha = 0.5$ was given to the two calculation methods:

$$\text{fPAR}(x,t) = \alpha \text{fPAR}_{\text{NDVI}} + (1 - \alpha)\text{fPAR}_{\text{SR}},\tag{11}$$

### 3.3. Improved Estimation of Light Use Efficiency

Ideally, plant vegetation has maximum light use efficiency but, in reality, the light use efficiency is affected by many external environmental factors: mainly temperature and precipitation. The light use efficiency thus was calculated by using the model established by Potter and Field [49,55] combined with the regional actual evapotranspiration model proposed by Zhou and Zhang in 1995 [57]. Specifically, the $\varepsilon$ is calculated as follows:

$$\varepsilon(x,t) = T_{\varepsilon 1}(x,t) \times T_{\varepsilon 2}(x,t) \times W_{\varepsilon}(x,t) \times \varepsilon^*,\tag{12}$$

$$T_{\varepsilon 1}(x,t) = 0.8 + 0.002 T_{opt}(x,t) - 0.0005 T^2_{opt}(x,t),\tag{13}$$

$$T_{\varepsilon 2}(x,t) = \frac{1.184}{\{1 + \exp[0.2(T_{opt}(x,t) - 10 - T(x,t))]\} \times \{1 + \exp[0.3(-T_{opt}(x,t) - 10 + T(x,t))]\}},\tag{14}$$

$$W_{\varepsilon}(x,t) = \left(\frac{1}{2}\right)\frac{EET(x,t)}{PET(x,t)} + \frac{1}{2},\tag{15}$$

where $T_{\varepsilon 1}(x,t)$ and $T_{\varepsilon 2}(x,t)$ are the temperature stress factors of the light use efficiency for pixel $x$ at time $t$, $W_{\varepsilon}(x,t)$ is the water stress factor of the light use efficiency for pixel $x$ at time $t$, $\varepsilon^*$ [g C MJ$^{-1}$] is the maximum light use efficiency under ideal conditions, $T(x,t)$ is the average temperature for pixel $x$ at time $t$, $T_{opt}(x,t)$ is the appropriate temperature for plant growth for pixel $x$ at time $t$; $EET(x,t)$ [mm] is the actual evapotranspiration and based on using Equation (16), which in turn depends on the Penman Equation (17) [44]. $PET(x,t)$ [mm] is the potential evapotranspiration, which is calculated as follows:

$$EET(x,t) = \frac{\{P(x,t) \times R_n(x,t) \times [P^2(x,t) + R^2_n(x,t) + P(x,t) \times R_n(x,t)]\}}{\{[P(x,t) + R_n(x,t)] \times [P^2(x,t) + R^2_n(x,t)]\}}\tag{16}$$

$$R_n(x,t) = [E_{p0}(x,t) \times P(x,t)]^{\frac{1}{2}} \times \{0.369 + 0.598 \times [\frac{E_{p0}(x,t)}{P(x,t)}]^{\frac{1}{2}}\} \tag{17}$$

$$PET(x,t) = \frac{E_{po}(x,t) + EET(x,t)}{2} \tag{18}$$

where, $P(x,t)$ [mm] is the precipitation for pixel $x$ at time $t$, $R_n(x,t)$ [MJ m$^{-2}$] is the net radiation for pixel $x$ at time $t$, $E_{p0}(x,t)$ is the local potential evapotranspiration, which is related to the temperature for pixel $x$ at time $t$, and Reference [58] is referenced to calculate $E_{p0}(x,t)$ based on vegetation-climate method established by Thornthwaite.

An important parameter of the CASA model is $\varepsilon^*$, the ideal-condition maximum light use efficiency, which can be assigned a fixed value when processing data corresponding to a small area [55,59]. $\varepsilon^*$ directly affects the value calculated for the light use efficiency. Without distinguishing vegetation types and geographical types, Field et al. [55] used 0.389 g C MJ$^{-1}$ as the global $\varepsilon^*$ of vegetation. Hunt [60] assigned an upper limit of 3.5 g C MJ$^{-1}$ to the $\varepsilon$, and Zhu et al. [56] estimated the $\varepsilon^*$ for various typical types of vegetation in China based on NPP estimated by remote sensing models. Goetz et al. [61,62] claimed that the $\varepsilon^*$ ranges from 0.42 to 3.8 g C MJ$^{-1}$, whereas, other results suggested that the $\varepsilon^*$ of wheat ranges from 1.46 to 2.93 g C MJ$^{-1}$ [63,64]. Different types of vegetation have different ecological and physiological structures, so their $\varepsilon^*$ should differ. Since the present study focuses on winter wheat, the environmental conditions of the study area along with the results of previous research are considered to fix the $\varepsilon^*$ of winter wheat at ranges from 0.42 to 2.93 g C MJ$^{-1}$. Then, the average of the range was calculated and rounded to get $\varepsilon^*$ of 1.7 g C MJ$^{-1}$, which had a good result for the improved CASA model in this study.

### 3.4. NPP-Yield Conversion Model

Winter wheat yield is the grain harvest of winter wheat, which often refers to the economic yield of winter wheat. Along with Ren et al. [65], the method was used to estimate winter wheat yield, which is

$$\text{Yield} = \frac{\alpha \sum \text{NPP} \times p \times \text{HI}}{1 - \omega} \times 10^{-2} \tag{19}$$

where Yield is the winter wheat yield [t ha$^{-1}$], $\alpha$ is the carbon-conversion coefficient (the carbon content of winter wheat is about 45%, so the $\alpha = 2.22$ [66]), $\sum \text{NPP}$ is the cumulative net primary production of organic matter over the entire growing season of winter wheat, $p = 0.9$ is the distribution coefficient of aboveground parts, $\omega = 12.5\%$ is the water-content coefficient of wheat grain during the storage period after harvest [67], and HI = 0.45 is the harvest index (this value is based on the growth characteristics of winter wheat in the Tongzhou and Shunyi districts of Beijing, as determined by previous research), $10^{-2}$ is the conversion factor between the unit g m$^{-2}$ and t ha$^{-1}$.

## 4. Results and Analysis

With the gradual increase of the hours of sunshine, temperature and precipitation; the winter wheat enters a stage of rapid growth, especially from March to May, which is a critical period for accumulating dry matter quality. Although this study analyzes the distribution of winter wheat yield over the entire growing season, the parameters fPAR, the light use efficiency, and the NPP from March to May are focused on to parametrize the improved CASA model.

### 4.1. Fraction of Absorbed Photosynthetically Active Radiation

The monthly NDVI and SR maximum and minimum values need to be determined before calculating fPAR based on Equation (11), which are listed in Table 1.

**Table 1.** NDVI and SR maximum and minimum values for winter wheat for each month of the growing season.

| Vegetation Indices | Maximum and Minimum | Months [3] | | | | | | | | |
|---|---|---|---|---|---|---|---|---|---|---|
| | | **10** | **11** | **12** | **1** | **2** | **3** | **4** | **5** | **6** |
| **NDVI** [1] | MAX | 0.557 | 0.684 | 0.687 | 0.548 | 0.430 | 0.493 | 0.757 | 0.854 | 0.687 |
| | MIN | 0.254 | 0.265 | 0.210 | 0.224 | 0.179 | 0.190 | 0.246 | 0.433 | 0.239 |
| **SR** [2] | MAX | 3.519 | 5.337 | 5.386 | 3.426 | 2.512 | 2.943 | 7.236 | 12.665 | 5.395 |
| | MIN | 1.681 | 1.720 | 1.531 | 1.578 | 1.437 | 1.469 | 1.651 | 2.528 | 1.629 |

[1] The full name of NDVI is Normalized Difference Vegetation Index; [2] The full name of SR is Ratio Vegetation Index; [3] "Months" refers to the winter wheat growing season from October 2008 to June 2009.

Figure 3 shows the results for fPAR from March to May 2009. The fPAR results for March, which average 0.47, were less than those for April and May, which fell mostly between 0.32 and 0.57. The fPAR for winter wheat in April continued to increase due to the work of photosynthesis and generally exceeds 0.40. The process of photosynthesis of winter wheat gradually weakened as May approaches, so the fPAR values concentrated in the range 0.40–0.60. The greatest average fPAR of 0.59 occurred in April 2009, whereas the average fPAR was 0.53 in May of the same year. These results reflected the mild weather during April, in which hydrothermal and sunlight conditions were suitable for winter wheat growth, resulting in increased photosynthesis of winter wheat and the subsequent increase in growth. In addition, the April and May fPAR results for Tongzhou and Shunyi districts did not differ significantly, which meant that the geographical environment of these two districts was suitable for winter wheat growth.

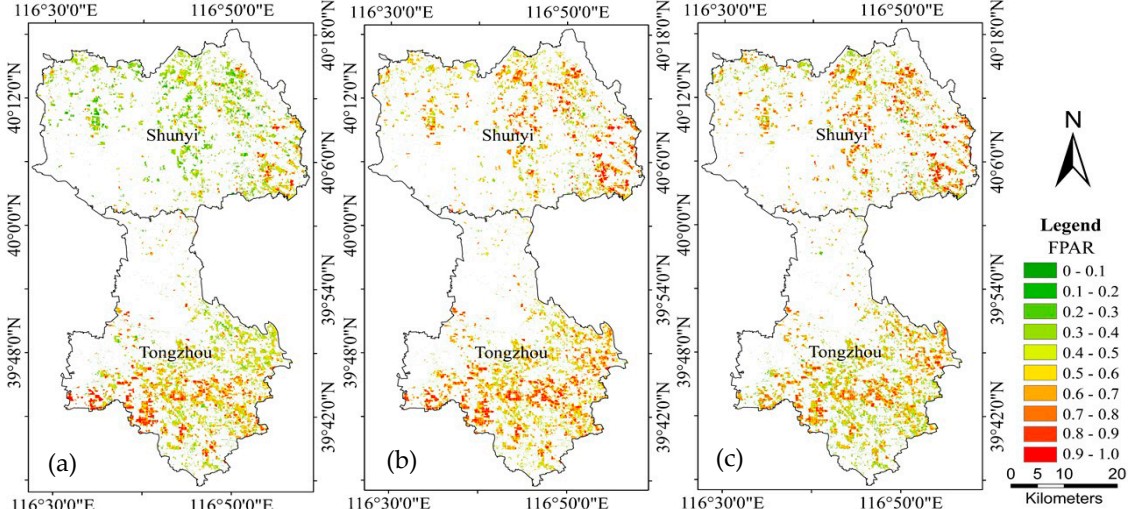

**Figure 3.** Spatial distribution of winter wheat fPAR from March to May, 2009: (**a**) March; (**b**) April; (**c**) May.

### 4.2. Light Use Efficiency

In 2009, the light use efficiency for winter wheat in the Tongzhou and Shunyi districts of Beijing remained relatively constant over each given month, although slight differences appeared due to differences in climate and environment. As shown in Figure 4, the $\varepsilon$ in March ranged from 0.67 to 0.71 g C MJ$^{-1}$, and the average monthly light use efficiency was 0.69 g C MJ$^{-1}$. In April, the $\varepsilon$ increased to reach 0.80–1.0 g C MJ$^{-1}$, and the average monthly $\varepsilon$ was 0.92 g C MJ$^{-1}$. Environmental conditions such as sunshine, temperature, and precipitation in May provided improved growing conditions for winter wheat, so the $\varepsilon$ in May ranged from 1.06 to 1.15 g C MJ$^{-1}$, with an average of 1.11 g C MJ$^{-1}$. Figure 4 also shows that the $\varepsilon$ in March and April was approximately the same in Tongzhou District as

in Shunyi District. In May, however, the $\varepsilon$ in Shunyi District was significantly greater than in Tongzhou District. This result reflected the better environmental conditions for winter wheat growth, such as temperature and precipitation, in Shunyi District from March to May, which allowed superior growth and development of winter wheat.

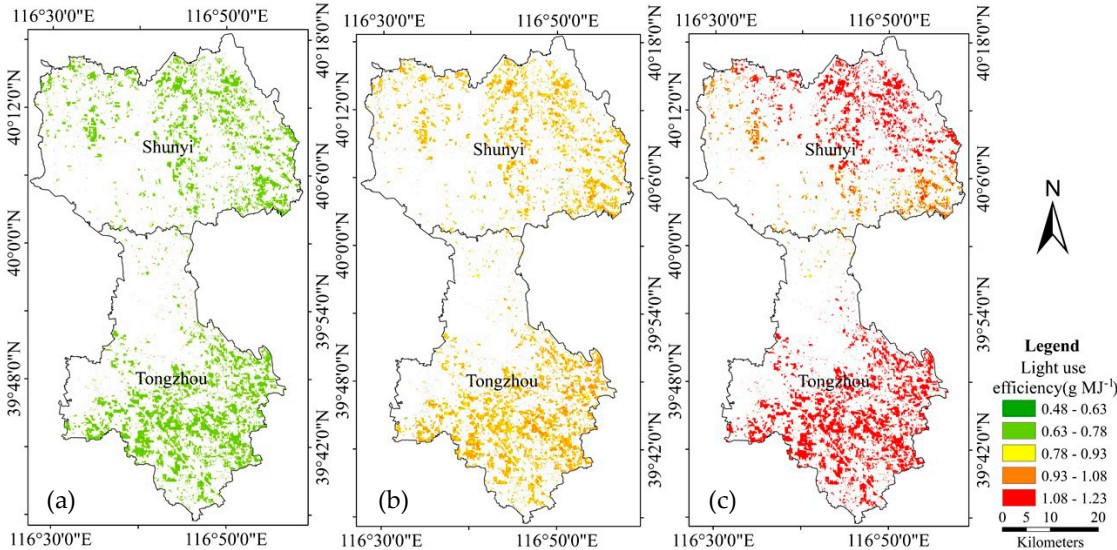

**Figure 4.** Spatial distribution of light use efficiency of winter wheat from March to May 2009: (**a**) March; (**b**) April; (**c**) May.

*4.3. Net Primary Production*

Figure 5 shows the spatial distribution of NPP in the study area from March to May, 2009. The NPP generally increased from March to May. In March, winter wheat gradually enters the greening stage and the erecting stage. The results for NPP in March were all less than 100 g C m$^{-2}$, and the average NPP was 25 g C m$^{-2}$. In April, most of the winter wheat was in the jointing stage, booting stage, and heading stage and the growth rate continued to accelerate. The NPP was mainly concentrated in the range 50–150 g C m$^{-2}$, with an average of 88 g C m$^{-2}$. In May, the average NPP reached a maximum of 128 g C m$^{-2}$ and mostly remained above 100 g C m$^{-2}$. The winter wheat growth increased continuously, and the dry matter quality accumulated rapidly.

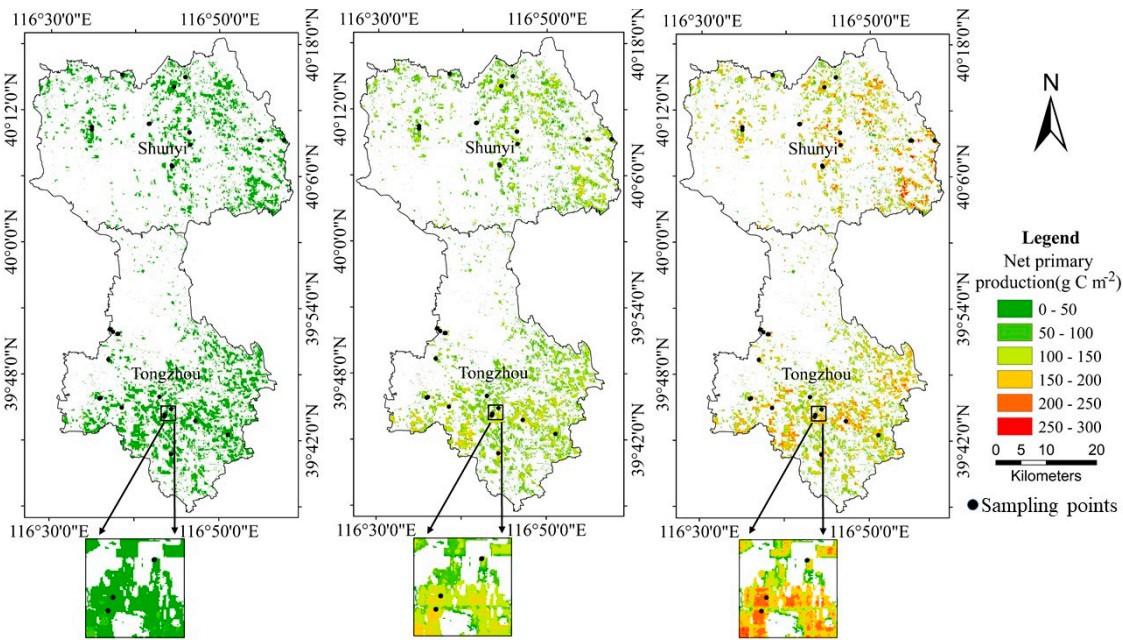

**Figure 5.** Spatial distribution of winter wheat net primary production (NPP) from March to May, 2009: (**a**) March; (**b**) April; (**c**) May.

The Tongzhou and Shunyi districts in Beijing have typical continental monsoon climates. The seasonal climate directly affects the formation of winter wheat NPP. Figure 6 shows the mean monthly NPP of winter wheat in the study area as a function of time over the entire growing season. From October to December 2008, the winter wheat NPP in the seedling stage decreased monotonically, primarily because, as the study area gradually entered the winter season, climatic factors such as solar radiation, temperature, and precipitation reached their low point. Thus, winter wheat growth was limited; the growth rate was slow, and the rate of NPP formation was relatively low. From January to March 2009, with the improvement of climatic conditions, the winter wheat gradually passed through the over-wintering stage and entered the stage of rapid growth and development, so the NPP began to increase. From April to June 2009, the winter wheat leaf area continued to increase, photosynthesis, respiration, and other growth activities continued to increase, and dry matter quality accumulated rapidly. Finally, in May, the winter wheat growth and NPP peaked.

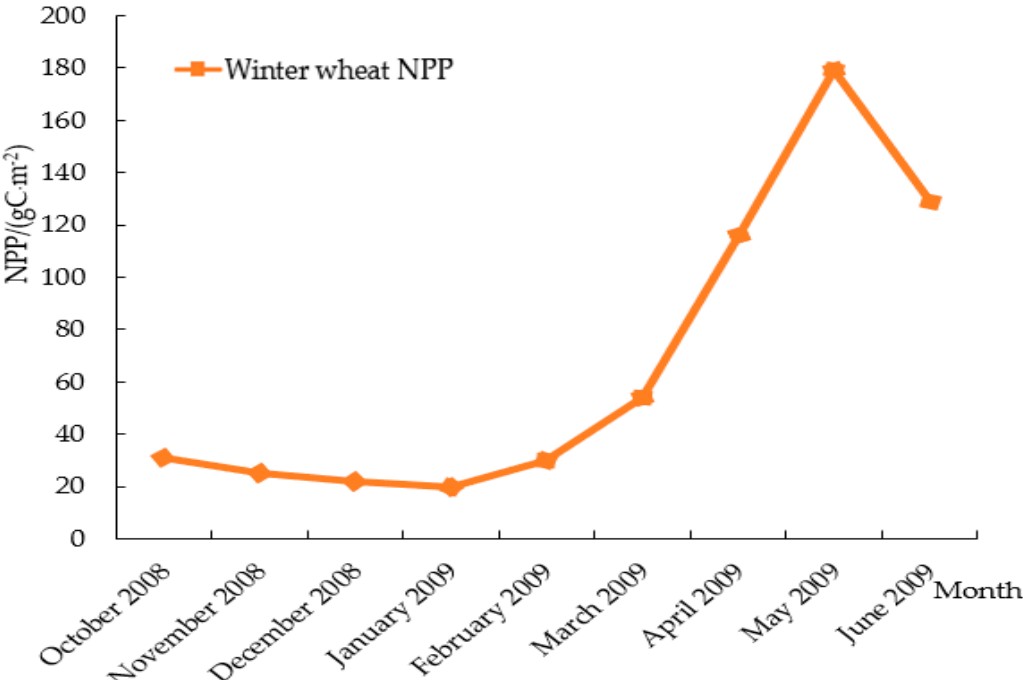

**Figure 6.** Mean winter wheat NPP as a function of time.

Figure 7 shows the spatial distribution of the total winter wheat NPP for the entire growing season, which was obtained by summing the NPP of each month. The overall distribution of winter wheat NPP in Tongzhou District was similar to that in Shunyi District and was concentrated in the range 300–700 g C m$^{-2}$, with an average NPP of 464 g C m$^{-2}$.

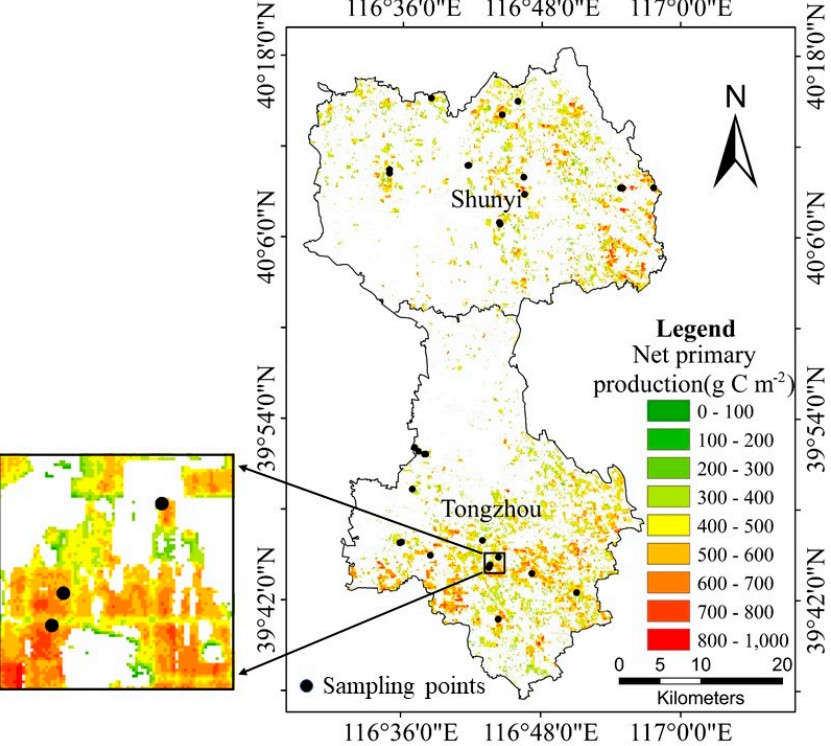

**Figure 7.** Spatial distribution of winter wheat NPP averaged for the 2008–2009 growing season.

### 4.4. Estimation of Winter Wheat Yield

In this study, the winter wheat NPP was used to estimate the winter wheat economic yield. First, the improved CASA model was applied to estimate the winter wheat NPP in the Tongzhou and Shunyi districts of Beijing for the 2008–2009 growing season. Next, the NPP-yield conversion model [Equation (19)] was used to convert the winter wheat NPP into the winter wheat economic yield.

Figure 8 shows the spatial distribution of the estimated winter wheat yield in the study area, which ranged from 3 to 8 t ha$^{-1}$ over more than 90% of the total study area. The output of the entire study area was about 50% for the range 3–5 t ha$^{-1}$ and 35% for the range 5–7 t ha$^{-1}$. The results show that, during the 2008–2009 growing season, the Tongzhou and Shunyi districts of Beijing had similar winter wheat growth trends, however, the winter wheat yield in the central part of the Tongzhou District and in the western part of the Shunyi District was higher than that in the northern part of the Tongzhou District and in the western part of the Shunyi District. Overall, the growth of winter wheat in the study area was ideal.

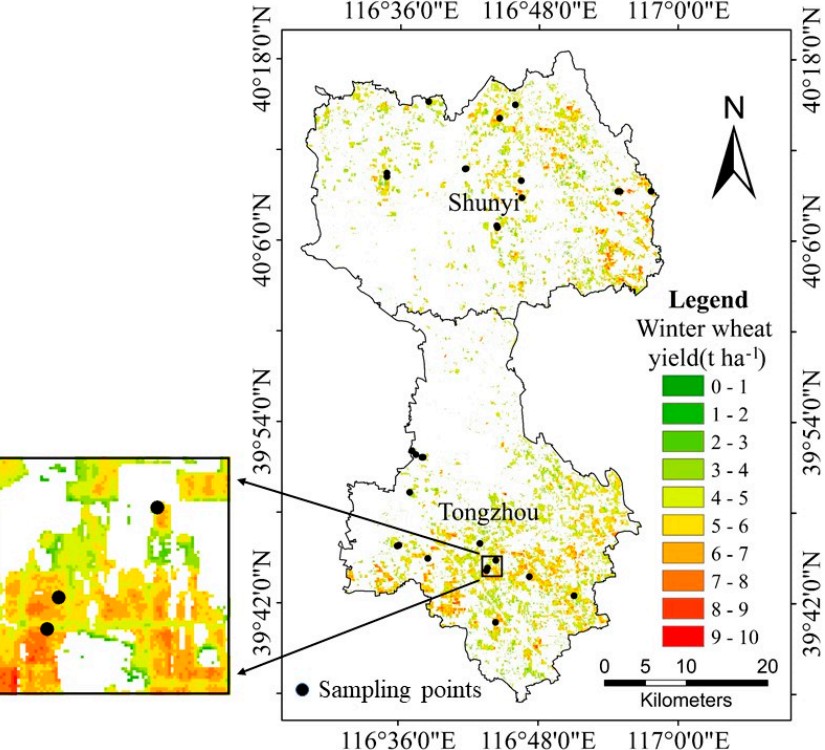

**Figure 8.** Spatial distribution of estimated winter wheat yield for the 2008–2009 growing season.

### 4.5. Verification of Estimated Yield

A total of 29 sampling points were selected to verify that the improved CASA model and the NPP-yield conversion model provided accurate estimates of winter wheat yield. Each pixel for the sampling points was averaged with the eight adjacent pixels to obtain the estimated yield for the given sampling point. Figure 9 shows the estimated yield as a function of the measured yield.

The determination coefficient $R^2 = 0.56$ and RMSE was 1.22 t ha$^{-1}$. This analysis gave an average absolute error of −0.57 t ha$^{-1}$ and an average relative error of −6.01%, which satisfied the accuracy requirements for estimating regional winter wheat yield. Thus, the proposed method may have good application potential.

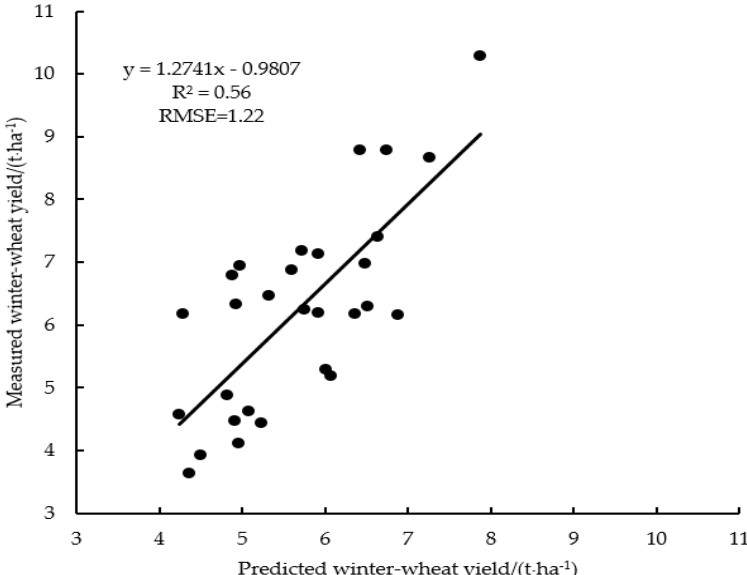

**Figure 9.** Predicted winter wheat yield as a function of measured winter wheat yield for the entire study area for the 2008–2009 growing season.

## 5. Discussion

Accurate and timely monitoring and evaluation of regional grain crop yield is a research priority of numerous countries, and remote sensing technology is widely used in a wide range of techniques to estimate crop yield [68–70]. MODIS remote sensing images are often used as a data source in many studies. These images may be downloaded for free and offer multiple data bands and a fast update frequency [71,72]. However, the choice of spatial resolution for MODIS images is only 250, 500, and 1000 m. Thus, single MODIS remote sensing images with high temporal resolution and low spatial resolution are often applicable for large areas. The Chinese domestic satellite HJ-1A/B, however, offers a relatively high spatial resolution of 30 m, but the return-time period is inferior to that of MODIS. Therefore, to study the Tongzhou and Shunyi districts of Beijing, the improved CASA model was developed to estimate winter wheat NPP based on HJ-1A/B images with high temporal and medium spatial resolution. At the same time, MOD17A2H images were incorporated to fully exploit the advantages of each remote sensing data source and obtain the spatial distribution of winter wheat NPP with a high spatial and temporal resolution (5-day intervals over the entire winter wheat growing season). The results provide support for monitoring and evaluation of winter wheat yield.

Thus, by using the improved CASA model and the NPP-yield conversion model, the winter wheat yield in the study area was estimated based on remote sensing images. However, the parameters used in the two models were often affected by factors such as vegetation type, geographical conditions, climate, and environment. Therefore, based on previous studies, some of the model parameters were optimized based on the actual environmental conditions in the study area during the winter wheat growing season. The most influential parameters for the improved CASA model were fPAR and the light use efficiency ($\varepsilon$).

The original CASA model only establishes a linear relationship between fPAR and SR. In this study, the two vegetation indices NDVI and SR are used to calculate $fPAR_{NDVI}$ and $fPAR_{SR}$, respectively, with each weighted the same, to estimate fPAR. In addition, although most studies calculate the maximum and minimum NDVI by using the 95% lower side value ($NDVI_{max}$) and the 5% lower side value ($NDVI_{min}$) of the NDVI maximum probability distribution extracted from remote sensing images [56], respectively a single constant was assigned to the NDVI maximum and minimum monthly because of the long growing season of winter wheat and the large number of remote sensing images involved in the calculations. In view of the actual situation of winter wheat over the entire

growing season, this approach was justified because it reduced the parameter error and improves the estimation accuracy.

The original CASA model used a soil-moisture model to calculate the water stress factor that appeared in the light use efficiency ($\varepsilon$). The maximum light use efficiency ($\varepsilon^*$) was assigned a value of 0.389 g C MJ$^{-1}$. In addition, many studies have been done to optimize the parameters of the model Reference [73], which showed that remote sensing and optimization techniques can be used to improve the precision of crop yield estimation, where the water stress factor depends on energy balance equation and $\varepsilon^*$ is equal to 0.29 g C/MJ for C3 crops. However, both the structure of the soil-moisture model and energy balance equation are complex and the parameters are not easy to determine, so $\varepsilon$ in this study was calculated by using the actual regional evapotranspiration model proposed by Zhou et al. [57], where temperature and precipitation can be used as input data to calculate parameters: the net radiation ($R_n$) and the local potential evapotranspiration ($E_{p0}$).

The value of $\varepsilon^*$ has a great influence on the simulation of NPP, and its size is controversial [55]. Therefore, designing experiment is the good way to determine the value of $\varepsilon^*$. Li et al. [74] applied Particle Swarm Optimizer (PSO) to search for the optimization of LUE retrievals through the CASA model combined with time-series NDVI and NPP ground measurements, which not only proved the significant difference of among various vegetation types, but also determined the specific value of $\varepsilon^*$ in grassland. However, the objective of this study was to estimate NPP of winter wheat based on the improved CASA model, so no specific experiments were designed to determine the value of $\varepsilon^*$. Upon combining this with the growth characteristics of winter wheat in the study area, the range of maximum light use efficiency ($\varepsilon^*$) was determined. Then, the average of the range was calculated and rounded to get $\varepsilon^*$ of 1.7 g C MJ$^{-1}$.

This study develops the improved CASA model and combines it with the NPP to estimate winter wheat yield from time-series satellite remote sensing images. The comparison of the estimated results with the ground-measured-yield data shows that the estimation accuracy satisfies the requirements for estimating regional winter wheat. Compared with the study by Ren et al. [65], both methods have good results, but this model has obvious advantages in particular, widely used remote sensing images, NPP distribution with high temporal and spatial resolution, and simpler input parameters. It is still an important objective to improve the accuracy of the estimated yield of crop, which can be considered from two aspects. One is the optimization of data sources. The estimation of crop yield depends partly on the quality and quantity of the input data. In this study, the mature integrated GPP product (MOD17A2H) was applied, which existed objectively and was widely used. Reference [73] creates an optimization model to compensate for missing data, which can improve the precision of crop yield estimation combined with Monteith model (CASA). Although the study object is not winter wheat, it is still worth paying attention to this optimization method. The other is the essential parameters in the model for estimating winter wheat yield based on remote sensing images: the $\varepsilon^*$ and the HI, which, based on previous research, were assigned constant values in the present study. However, these two parameters are affected by other factors, so variable processing must be done in addition to relevant experimentation, which will be the subject of future research.

A total of 29 ground-measured winter wheat yield data were used in this study and winter wheat yield were concentrated in the relatively small range of 3–8 t ha$^{-1}$. An ideal result was obtained by using 29 sampling points in this study area, but if the study area scale is continuously expanded, further research is required to verify the accuracy and applicability of the proposed model.

## 6. Conclusions

This study combined two kinds of satellite data: HJ-1A/B remote sensing images and MOD17A2H data products as input for an improved CASA model to estimate the net primary production (NPP) during the entire 2008–2009 winter wheat growing season. The NPP-yield conversion model was then used to estimate winter wheat yield on a regional scale. The results give a determination coefficient $R^2$ = 0.56 between the measured and estimated winter wheat yield, RMSE of 1.22 t ha$^{-1}$, and an average

relative error of −6.01%. Thus, the improved CASA model combined with the NPP-yield conversion model satisfied the requirements for regional-scale estimates of winter wheat yield from remote sensing images. This approach may thus be applied in the field and can also provide a benchmark for a wide range of models for estimating crop yield.

**Author Contributions:** Y.W.; processed and analyzed the data and drafted the manuscript. X.X.; guided the ideal, designed the experiment, advised on data analysis and revised the manuscript. G.Y.; L.H.; L.F.; P.W. and G.C. were involved in the experiments, ground data collection, and manuscript revision. All authors read and approved the final version.

**Funding:** This work was supported by the National Natural Science Foundation of China (Grant No. 41571416) and the National Key Research and Development Program (Grant No. 2017YFD0201501).

**Acknowledgments:** We appreciate Hong Chang and Weiguo Li for their help during field data collection. The funders had no role in choosing the study design or in the collection, analysis and interpretation of the data, in the writing of the report, or in the decision to submit the article for publication.

**Conflicts of Interest:** The authors declare no conflict of interest.

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
