# Peer review of "An Improved CASA Model for Estimating Winter Wheat Yield from Remote Sensing Images"

_remotesensing, doi:10.3390/rs11091088_

Round 1

Reviewer 1 Report

The research article "An Improved CASA Model for Estimating Winter Wheat Yield from Remote Sensing Images" is an interesting one because it introduces new improvement to CASA model related to APAR and light use efficiency. It also uses a different and advanced satellites HJ-1A/B for estimating NPP instead of using Landsat which is an interesting issue.

However the following major problems must be fixed in order to improve the readability of the article:

1-Authors should include in the introduction  research studies about using Energy balance equations, remote sensing  and optimization techniques to estimate crops such as the following

Toward Precision in Crop Yield Estimation Using Remote Sensing and Optimization Techniques, MDPI Journal of Agriculture, Vol. 9, No. 3, 54; doi: 10.3390/agriculture9030054, 2019.

2-When speaking in the introduction about using HJ-1A/B and MOD17A2H add few words to introduce them. Although they are explained and data and methods it is worth adding few words such as an example multispectral  two satellites HJ-1A/B high  temporal and medium spatial resolutions. Another MOD17A2H is the Gross Primary Production (GPP) from Modis with 500 meter spatial resolution.

3- What is the difference between NPP and GPP add some explanation in the text.

4-Authors indicate in Lines 140 to 141 that MODIS GPP was used to compensate for lack of data in HJ-1A/B. The problem is in the large difference in spatial resolutions one is 30 meters the other is 500 meters 

In addition, Modis data have 8 days temporal resolution compared from 2 to 5 days for HJ. Another issue how did you convert Modis MOD17A2H GPP to NPP?

5-No need to add the location in line 96 to 97 if the map is added in figure 1.

6-No need for figure 1c. the solution is to enlarge 1b and add sampling points in it.

7-references are needed for equations 4 to 7.

8-If samples number is larger than 29 this could give more credibility to the research work. Please justify why the samples are not large enough compared to the size of the area of study.

9-References are needed for equations 4 to 7.

10-In equation 8, there is a concern  about case NDVI is negative what would be the value of fpar?

In addition, what would be the case for SR if NDVI(x,t) is equal to 1?

11-Light use efficiency is calculated differently in many papers such as 

A- Estimating the Maximal Light Use Efficiency for Different Vegetation through the CASA Model Combined with

Time-Series Remote Sensing Data and Ground Measurements, MDPI Remote Sens. 2012, 4, 3857-3876; doi:10.3390/rs4123857 

B- Toward Precision in Crop Yield Estimation Using Remote Sensing and Optimization Techniques, MDPI Journal of Agriculture, Vol. 9, No. 3, 54; doi: 10.3390/agriculture9030054, 2019.

In the last one paper water stress factor  is calculated differently 

Discuss these papers and explain how the ideal-condition maximum light use efficiency was selected to be 1.7 g C/MJ. 

12-Why alpha, p, and HI (carbon content, water content, and harvest index) in Equation 16 are given these specific constant values based on what criteria? Explain more about how these values are selected. In addition, do these values change during the crop growth stages?

13- It is better to add a crop map showing wheat crop only.

14- The maps in Figure 4, 6 and 7 need to be more clearer I would suggest to create a zoom in image  beside each map showing specific area where intensive verification is done with points.

Author Response

Cover letter

Dear Reviewer,
Thank you very much for your reply and reviewers’ constructive suggestions for improving our manuscript. We revised the manuscript point by point very carefully according to the reviewers’ suggestions. Attached please find the revised manuscript and the following responses to reviewers’ comments. In the manuscript, all revised parts are highlighted in red for convenience of being reviewed by you and reviewers.

With best regards,
Co-authors

Reviewer 2 Report

I have reviewed the manuscript remotesensing-489168 and I believe the authors have carried out an interesting study to estimate winter wheat yield from remote sensing images by improving a developed model (CASA).  

I consider that there are some details that should be improved in order to a better understanding of the manuscript. I have some suggestions to improve it. My comments are detailed below:

1.      Throughout the entire manuscript, authors used “d” as the unit of day. Nevertheless, as the time unit of the international system is second (s), d as day can be confused. I suggest to use the entire word and then the abbreviation “day (d)” in the abstract section and the first time this word is used in the introduction section.

2.      L 18-19: Net Primary Production (NPP) need a blank space.

3.      L 17-22. Very long sentence. Try to split it.

4.      In different lines of the manuscript, there are references used as part of a sentence and the name of the first author is indicated in the sentence. Nevertheless, the number of the sentence is not right after the author, but at the end of the sentence. The number should be placed after the name (eg. L 50: … the work of Xu et al. [20], who used…).  Be careful with L 2008-210, where the authors referred to Ruimy et al., but at the end of the sentence there are tree references and only one is from Ruimy.

5.      L 69-70. Some words are repeated.

6.      L 69-70. Why do not the authors indicate the acronym of photosynthetically active radiation and light use efficiency in this sentence? The acronyms of fraction photosynthetically active radiation (fPAR) and even the maximum light use efficiency (ε*) are included in the same paragraph (L74 and 76).

7.      L85-91. In the last paragraph of the Introduction section, the objective of the study should be explained. Nevertheless, in this manuscript, authors have summarized the methodology carried out.

8.      L 96-107. In those paragraphs, there are descriptions of the studies areas (Tongzho and Shunyi Districts). Nevertheless, the description of both Districts are not the same. While in Tongzho District a description of climate and meteorological data can be read, in Shunyi District, this information is inexistent.

9.      In the description of the satellite images (L11- 149), only the spatial resolution was mentioned of both satellites and the temporal resolution of HJ-1A/B. Please, describe better both satellites characteristics with the spectral and radiometric resolutions of both and temporal resolution of MODIS. Not all readers know the characteristics of all satellites.

10.  I have doubts about the influence of MODIS information in the study. Can both types of image be used in the same method without a great variation of the results?. Only considering the spatial resolution of both images (30 m and 500 m), the differences are obvious. Nevertheless, are the spectral range of both, red and NIR bands, similar in order to compare the vegetation indexes?

11.  L129-132. “The winter wheat growing season” is repeated tree time in the same sentence.

12.  L 133-134: Revise the sentence. Should be …”Savitzky-Golay filter was applied to…?

13.  L 189-190. Revise the sentence. Although it is correct, it can be confusing. A suggestion: “The direct calculation of the total solar radiation was not included in this study because the small number of meteorological stations that provide total solar radiation”.

14.  L 222-224. Authors use “which” twice in the same sentence and one more in the next sentence. Revise the sentence.

15.  Are the vegetation indices (Table 1) results of the study? I would include this information in the Result Section of the manuscript.

16.  Table 1: Although the information was previously explained in the manuscript, each table must be read independently of the text. Include a footnote with the description of NDVI and SR.

17.  Figures 2, 3 and 4. The captions of the figures can be simplified. (eg. Figure 2: Spatial distribution of winter wheat fPA from March to May 2009: (a) March; (b) April; (c) May).  

18.  L 304. Authors repeat “average” twice in this sentence, and I ask if the first time was a mistake.

19.  L 325-327. This information should be included in the 2.1 Study Area Section.

20.  L 350-354. This information is about the parameter used in the model and were included in the 3.4 NPP-Yield Conversion Model Section. This information is no necessary in the Result Section.

21.  L 366. When a sentence begins with a number, it is better to put it with letters.

22.  There is no discussion of the results. Throughout the Discussion Section, a summarize of all the study was presented but there is no discussion of the results obtained. Only, L 417- 420 indicate that the estimation accuracy satisfies the requirements for regional winter wheat, which is consistent with previous studies. I expected a deeper analysis of the results obtained for similar purposes but with other techniques, especially with the unmodified (original) CASA method if exists. I encourage the authors to improve this section to defend that a R2 = 0.56, a RMS of 1.22 t ha-1 and an average relative error of -6.01% satisfy the requirements for regional-scale estimates.

23.  L 384. I am not sure that MODIS data can not be used at the country level. Do the authors have any reference to say that?

24.  Revise denominators in units in the entire manuscript. (Eg: t/ha should be t ha-1).

Author Response

(The authors gave the same response as above.)

Round 2

Reviewer 1 Report

The authors have implemented and corrected most of the issues required by the reviewers except few minor issues.

1-Some minor English language editing is required 

2-It is important that the authors reverse the color of the legends for the maps of the NPP and wheat yield such that green becomes  for high yield and red for low.

3- If possible to reduce the number of equations and ease the task of reading the of the manuscript by putting the equations in an appendix with and the same for the term and variables explanation.

Reviewer 2 Report

Thanks for the efforts to improve the manuscript.